# Bedside Risk Scoring for Carbapenem-Resistant Gram-Negative Bacterial Infections in Patients with Hematological Malignancies

**DOI:** 10.3390/idr17040092

**Published:** 2025-08-01

**Authors:** Sare Merve Başağa, Ayşegül Ulu Kılıç, Zeynep Ture, Gökmen Zararsız, Serra İlayda Yerlitaş

**Affiliations:** 1Department of Infectious Diseases, Eyüp Sultan State Hospital, İstanbul 34654, Türkiye; saremerve@erciyes.edu.tr; 2Department of Infectious Diseases, Faculty of Medicine, Erciyes University, Kayseri 38039, Türkiye; aulukilic@erciyes.edu.tr; 3Private Gürlife Hospital, Infectious Diseases Clinic, Eskişehir 26010, Türkiye; 4Department of Biostatistics, School of Medicine, Erciyes University, Kayseri 38039, Türkiye; gokmenzararsiz@erciyes.edu.tr (G.Z.); serrayerlitas@erciyes.edu.tr (S.İ.Y.)

**Keywords:** carbapenem resistance score, hematological malignancies, carbapenem resistance, Gram-negative infections, bedside score

## Abstract

Background/Objectives: This study aimed to create a ‘carbapenem resistance score’ with the risk factors of carbapenem-resistant Gram-negative bacterial infections (GNBIs) in patients with hematological malignancies. Methods: Patients with carbapenem-resistant and susceptible GNBIs were included in this study and compared in terms of risk factors. Three models of “carbapenem resistance risk scores” were created with statistically significant variables. Results: The study included 154 patients with hospital-acquired GNBIs, of whom 64 had carbapenem-resistant GNBIs and 90 had carbapenem-susceptible GNBIs. Univariate and multivariate analyses identified several statistically significant risk factors for carbapenem resistance, including transfer from another hospital or clinic (*p* = 0.038), prior use of antibiotics like fluoroquinolones (*p* = 0.009) and carbapenems (*p* = 0.001), a history of carbapenem-resistant infection in the last six months (*p* < 0.001), rectal *Klebsiella pneumoniae* colonization (*p* < 0.001), hospitalization for ≥30 days (*p* = 0.001), and the presence of a urinary catheter (*p* = 0.002). Notably, the 14-day mortality rate was significantly higher in the carbapenem-resistant group (*p* < 0.001). Based on these findings, three risk-scoring models were developed. Common factors in all three models were fluoroquinolone use in the last six months, rectal *K. pneumoniae* colonization, and the presence of a urinary catheter. The fourth variable was transfer from another hospital (Model 1), a history of carbapenem-resistant infection (Model 2), or hospitalization for ≥30 days (Model 3). All models demonstrated strong discriminative power (AUC for Model 1: 0.830, Model 2: 0.826, Model 3: 0.831). For all three models, a cutoff value of >2.5 was adopted as the threshold to identify patients at high risk for carbapenem resistance, a value which yielded high positive and negative predictive values. Conclusions: This study successfully developed three practical risk-scoring models to predict carbapenem resistance in patients with hematological malignancies using common clinical risk factors. A cutoff score of >2.5 proved to be a reliable threshold for identifying high-risk patients across all models, providing clinicians with a valuable tool to guide appropriate empirical antibiotic therapy.

## 1. Introduction

Patients with hematological malignancies face a significantly elevated risk of Gram-negative bacterial infections (GNBIs). This increased susceptibility is primarily attributed to gastrointestinal mucositis induced by chemotherapy and extended periods of neutropenia [1]. Unfortunately, recent years have witnessed a global surge in bacterial resistance to carbapenem antibiotics, leading to increased mortality rates within this patient demographic [2]. Compounding this issue is the absence of established antibiotic regimens for carbapenem-resistant GNBIs in routine febrile neutropenia protocols. Consequently, there is a pressing need to identify risk factors for carbapenem resistance, allowing for early empirical therapy guidance and improved patient outcomes.

The primary objective of this study is to ascertain the risk factors associated with carbapenem-resistant GNBIs in patients with hematological malignancies. Our secondary goal is to develop a bedside scoring system that can identify high-risk patients. This scoring system, referred to as the ‘carbapenem resistance score’, aims to incorporate carbapenem-resistant factors into early, effective empirical treatment. By applying this scoring system, we seek to reduce mortality rates among patients at high risk of carbapenem-resistant Gram-negative bacterial infections (CR-GNBIs).

## 2. Materials and Methods

### 2.1. Setting

This study focused on patients with hematological malignancies who developed GNBIs at Erciyes University Hematology-Oncology Hospital and the Bone Marrow Transplantation and Stem Cell Treatment Center.

### 2.2. Study Design

#### 2.2.1. Determination of Risk Factors for Carbapenem-Resistant Gram-Negative Bacterial Infections and Data Collection

This study was designed to compare risk factors between hospitalized patients with carbapenem-resistant and carbapenem-susceptible GNBIs. Data collection utilized a combined design, with retrospective data gathered from January 2018 to January 2019, followed by prospective data collection (because of the insufficient amount of data) between February 2019 and February 2021. Demographic information, clinical characteristics, and details on treatments (such as steroid, blood products) and comorbidities were collected from patients files.

The eligible participants were patients older than 18 with hematological malignancies (such as acute leukemia, lymphoma, multiple myeloma, aplastic anemia, etc.) who developed GNBIs during their hospital stay. Patients were monitored for clinical infections, confirmed by Gram-negative bacterial growth in samples, and cases where this growth was considered, colonization were excluded. To ensure the independence of data points, only one infection episode per patient was included in the final analysis.

#### 2.2.2. Definitions

Cumulative Steroid Dose: The total steroid dose administered prior to the onset of bacteremia was calculated, converted into prednisolone equivalent doses (Appendix A), and adjusted per kilogram. This yielded a recorded mg/kg dose for each patient [3,4].Effective Empirical Treatment: The initial empirical treatment was deemed effective if a Gram-negative microorganism suspected as the potential cause of infection demonstrated in vitro sensitivity to the antibiotic administered on the day the clinical culture sample was taken.Effective Treatment: Patients were considered to receive effective treatment if they received at least one active antibiotic, based on in vitro susceptibility, against a pathogen suspected as the cause of infection.Carbapenem Resistance: The BD Phoenix automated system (Beckton Dickinson, Franklin Lakes, NJ, USA) was utilized for identifying the isolate and determining its antibiotic susceptibility. The Kirby–Bauer disk diffusion method was employed to confirm carbapenem resistance. Microorganisms moderately susceptible or resistant to meropenem, ertapenem, and imipenem were categorized as “carbapenem-resistant” according to culture and antimicrobial susceptibility test results.

#### 2.2.3. Creating a “Carbapenem Resistance Risk Score”

Multivariate analysis was conducted, incorporating all risk factors with a statistical significance level of *p* value (*p*) < 0.05 and those identified as risk factors in univariate analyses for carbapenem-resistant GNBIs. Three distinct “carbapenem resistance scores” were developed through a stepwise variable elimination process, utilizing variables that demonstrated statistical significance in univariate analyses for carbapenem resistance. This process resulted in three distinct scoring models, each composed of a total of four significant variables. To create a practical point system, the statistical coefficient (β) for each of these four variables was rounded to assign a simple score.

A threshold with high positive and negative predictive values across all three scoring models was deemed the marker for carbapenem resistance. Patients surpassing this threshold were categorized as being at a high risk for CR-GNBIs.

The discriminative power of each of the three final models was then evaluated using Receiver Operating Characteristic (ROC) curve analysis, and the Area Under the Curve (AUC) was calculated with a 95% confidence interval (CI). To determine a clinically useful threshold for identifying high-risk patients, different cutoff scores were evaluated. A final cutoff value was selected based on its ability to provide high positive and negative predictive values across all three models. Patients with a total score exceeding this established threshold were categorized as being at high risk for a carbapenem-resistant Gram-negative bacterial infection.

### 2.3. Statistical Analysis

The normality of the data was assessed through the Shapiro–Wilk test, histograms, and quantile–quantile (q-q) plots. Independent two-sample *t*-tests and Mann–Whitney U tests were employed to compare quantitative data within paired groups. For comparing categorical data, Fisher–Freeman-–Halton’s exact test, Pearson’s chi-square test, and Fisher’s exact chi-square test were utilized. Variables demonstrating statistical significance were integrated into the univariate logistic regression model for carbapenem resistance. The outcome, referred to as the “carbapenem resistance score,” was derived through a stepwise elimination process. The discriminative power of this score was evaluated using the Receiver Operating Characteristic (ROC) Area Under the Curve (AUC) and a 95% confidence interval (CI). Sensitivity, specificity, and positive and negative cutoff statistics were calculated, accompanied by 95% CIs for the cutoff values. Statistical analysis was performed using TURCOSA (Turcosa Analytics Ltd., Co., Kayseri, Turkey, www.turcosa.com.tr) statistical software. The threshold for statistical significance was set at *p* < 0.05.

## 3. Results

The study included 154 patients, with 64 in the carbapenem-resistant group and 90 in the carbapenem-susceptible group. While the distribution of age, gender, and specific hematological malignancies like acute leukemia (AL) and lymphoma did not show a statistically significant difference between the two groups, a notable finding was the higher rate of patient transfers from another hospital or clinic in the resistant group (15.6%) compared to the susceptible group (5.6%), a difference that was statistically significant (*p* = 0.038). The analysis of prior antibiotic use within the last six months revealed several significant risk factors for developing CR-GNBIs. Specifically, previous use of β-lactam/β-lactamase inhibitors (*p* = 0.001), fluoroquinolones (*p* = 0.009), carbapenems (*p* = 0.001), and colistin (*p* < 0.001) was significantly more common in all cases in the carbapenem-resistant cohort. Furthermore, a history of a carbapenem-resistant infection within the last six months (*p* < 0.001), a history of rectal colonization with *K. pneumoniae* (*p* < 0.001), and a history prior hospitalization in an ICU (*p* = 0.007) were also identified as significant predictors of resistance. The cumulative dose of steroids and the number of blood products (both platelets and erythrocytes) received were also significantly higher in the carbapenem-resistant group. In contrast, factors such as the presence and duration of neutropenia were not found to have a significant correlation with carbapenem resistance. Multivariate analyses revealed that rectal *K. pneumoniae* colonization significantly increased the risk for a CR-GNBI by 11 times, a history of colistin use increased the risk by 4 times, and a previous occurrence of a CR-GNBI increased the risk by 9 times. Furthermore, transfer from another hospital, use of fluoroquinolones or carbapenems, and a history of an ICU stay were also identified as independent risk factors for resistance (Table 1).

The analysis of hospitalization duration and invasive procedures, as detailed in Table 2, revealed significant risk factors for CR-GNBIs. Patients who developed CR-GNBIs had a significantly longer hospital stay before the onset of infection compared to the carbapenem-susceptible group (*p* < 0.001). This risk increased with the length of stay, as patients hospitalized for ≥30 days were 3.3 times more likely to develop a resistant infection (*p* = 0.001). Regarding invasive procedures, the presence of a urinary catheter was a prominent risk factor, increasing the odds of a CR-GNBI by 3.4 times (*p* = 0.002). Other procedures, including the use of a central venous catheter, hemodialysis, and surgical interventions, did not show a statistically significant association with carbapenem resistance. Among patients with carbapenem-resistant GNBIs, the most prevalent microorganism was *K. pneumoniae* (43.8%), while in patients with carbapenem-susceptible GNBIs, *Escherichia coli* accounted for the majority of cases (58.9%). Furthermore, it was noted that the 14-day mortality rate among patients with CR-GNBIs was significantly higher than that of patients with carbapenem-susceptible GNBIs (*p* < 0.001).

Three distinct models, each comprising four variables, were developed for the calculation of the carbapenem resistance risk score in patients with hematological malignancies. The variables common to all three models were urinary catheter use, fluoroquinolone use in the last 6 months, and the presence of rectal *K. pneumoniae* colonization during the infection. Model 1 incorporated being transferred from another hospital or service as the fourth variable, Model 2 included the presence of CR-GNBIs in the last 6 months, and Model 3 accounted for hospitalization lasting 30 days or more (Table 3).

The discriminative power of this score was assessed using the ROC AUC along with a 95% CI.

The AUC values for the models intended to predict carbapenem resistance were as follows:Model 1: AUC = 0.830, 95% CI (0.762–0.898; *p* < 0.001);Model 2: AUC = 0.826, 95% CI (0.759–0.893; *p* < 0.001);Model 3: AUC = 0.831, 95% CI (0.764–0.899; *p* < 0.001).

These AUC values indicate that the models exhibit strong discriminative power in predicting carbapenem resistance. A threshold value of >2.5, which yielded high positive and negative cutoff values for all three models, was adopted as the criterion for identifying the risk of carbapenem resistance. The sensitivity, specificity, and positive and negative cutoff values associated with this determined threshold are presented in Appendix A.

## 4. Discussion

Infections remain a primary cause of death in patients with hematological malignancies, with the landscape of causative pathogens showing a concerning shift from Gram-positive to Gram-negative bacteria [2,3,4]. The prognosis for these patients is further complicated by the alarming global rise in carbapenem resistance, which significantly increases mortality [3]. A critical issue is that current empirical treatment algorithms for febrile neutropenia do not adequately address these multi-drug resistant GNBIs, creating a gap in patient care [5]. Therefore, identifying patients at high risk for CR-GNBIs is crucial for guiding early, appropriate empirical treatment and improving outcomes. A readily accessible bedside scoring system, combined with targeted antibiotic strategies, could enable this necessary early intervention

Our study identified a history of fluoroquinolone, carbapenem, colistin, and β-lactam + β-lactamase inhibitor use, as well as the use of more than three antibiotics in the last six months, as risk factors for the development of carbapenem-resistant GNBIs. These findings are consistent with the current literature data [6,7,8]. Prolonged exposure to antibiotics can lead to bacterial resistance due to selective antimicrobial pressure. Antibiotics exert selective pressure on bacteria at both high and low concentrations, increasing the risk of CR-GNBIs. Antibiotics suppress susceptible bacterial groups and alter bacterial density, favoring bacteria with antibiotic resistance genes. This facilitates the dominance of resistant bacteria and the horizontal gene transfer of antibiotic resistance genes between different bacteria [9]. A meta-analysis revealed that exposure to carbapenems increased the risk fourfold and fluoroquinolone exposure increased the risk twofold for carbapenem-resistant *K. pneumoniae* infections [8]. A case–control study demonstrated that 91.6% of patients with hospital-acquired infections with carbapenem-resistant *K. pneumoniae* had a history of antibiotic use before reproduction [5]. Fluoroquinolones are commonly used antibiotics in chemotherapy and post-transplant prophylaxis for patients with hematological malignancies. Fluoroquinolones can induce resistance to both fluoroquinolones and carbapenems by up-regulating the multidrug efflux pump MexEF-OprN and downregulating the porin OprD, which contributes to carbapenem resistance [8]. A meta-analysis of 20 studies and 3715 cases demonstrated that fluoroquinolone exposure increased the risk of carbapenem-resistant *K. pneumoniae* infections by 1.75-fold [10]. A study group, including the European Conference on Infections in Leukemia group of researchers, re-evaluated fluoroquinolone prophylaxis during neutropenia in 2018 and recommended fluoroquinolone prophylaxis in areas with low or moderate fluoroquinolone resistance rates (<27%) [11]. Therefore, the routine use of fluoroquinolone prophylaxis should depend on national antimicrobial use policies and local epidemiological data.

Our study revealed that the use of colistin in the last six months increased the risk of carbapenem-resistant GNBIs by 4.1 times. A case–control study comparing 95 patients with carbapenem-resistant *K. pneumoniae* infection and 100 patients without resistant growth found that the use of colistin in the pre-infection period increased the risk of carbapenem-resistant GNBIs [6].

Colonization with carbapenem-resistant microorganisms has been reported as a significant risk factor for carbapenem-resistant bloodstream infections (BSIs) in patients with hematological malignancies [3,12,13]. In this group of patients diagnosed with rectal colonization with resistant bacteria, the risk of invasive infection increases due to disease-related immunosuppression, neutropenia, and impaired intestinal microbiota homeostasis resulting from broad-spectrum treatments [14]. In a study involving 225 patients with hematological malignancies found that 17 of 94 (18%) patients colonized with carbapenem-resistant isolates developed bloodstream infections (BSIs) due to carbapenem-resistant Enterobacteriaceae during their neutropenic period [13]. According to the results of a 3-year surveillance program of multiple-drug-resistant (MDR) GNBIs among 241 hematopoietic stem cell transplantation recipients in a transplant unit, the cumulative incidence of MDR GNBIs was 10.5%. Multivariate analysis reported that colonization with MDR GNBIs at the time of admission to the transplant unit increased the risk of resistant infections [15]. Our study demonstrated that rectal *K. pneumoniae* colonization increased the incidence of carbapenem-resistant GNBIs by 11 times. Therefore, it is essential to conduct colonization surveillance in this group of patients and other critically ill groups and consider colonization when selecting empirical treatment.

Transfer from another hospital or service was found to be a risk factor for carbapenem-resistant infections in our study. The increased risk of cross-contamination with antibiotic use, invasive procedures, and transplantation in different services or health institutions may be responsible for this. In a study in our country, colonization of carbapenem-resistant Gram-negative bacteria was observed in patients diagnosed with hematological malignancies, with patient transfer from another hospital increasing the colonization risk 6.5 times [16]. Patients with a history of transplantation from other services and hospitals have an increased risk of colonization with resistant microorganisms and a higher risk of invasive infection, especially in patients colonized with resistant microorganisms.

Our study found that hospitalization for 30 days or more before infection and a history of ICU hospitalization in the last six months tripled the risk of developing carbapenem-resistant GNBIs. The length of hospital stays, particularly in the ICUs, is closely associated with colonization of carbapenem-resistant *K. pneumoniae*, and the risk of both colonization and infection increases with the duration of ICUs stay [6]. Rates of colonization and infection with carbapenem-resistant Gram-negative bacteria are higher in the ICUs, which serves as a major hospital reservoir of MDR bacteria [10]. Screening for rectal colonization should be performed in patients transferred from the ICUs to detect resistant infections and prevent cross-contamination. Contact isolation precautions should be implemented in high-risk patients until screening results are available.

Furthermore, a history of carbapenem-resistant GNBIs in the last six months was found to increase the risk of CR-GNBIs ninefold. Satlin et al. reported that having a carbapenem-resistant Enterobacteriaceae infection in the three months before the current infection increased the risk of BSIs from carbapenem-resistant Enterobacteriaceae twelvefold [17].

The existing guidelines uniformly recommend initiating monotherapy with a beta-lactam that demonstrates activity against Pseudomonas aeruginosa (such as Piperacillin/tazobactam, imipenem, meropenem, cefepime, and ceftazidime) for high-risk patients with hematological malignancies. Combination therapy is advised for clinically unstable patients and when there is a suspicion of infection caused by resistant Gram-negative or Gram-positive bacteria. The guidelines for combination therapy in cases of resistant GNBIs are comparatively more restrictive compared to those for Gram-positive infections [18]. The increasing incidence of CR-GNBIs and the associated mortality rates underline the importance of identifying risk factors for carbapenem resistance. Despite numerous studies investigating risk factors for CR-GNBIs in response to the growing rates of resistance, there is currently no bedside scoring system to guide empirical treatment in patients with hematological malignancies. A number of studies have previously developed models or risk scores to predict various aspects of bacterial infections, such as the development of carbapenem resistance in specific pathogens. For instance, Yang et al. developed a model for predicting carbapenem resistance in *K. pneumoniae* infections, showcasing the influence of various factors like age, gender, cardiovascular disease, prolonged hospital stays, a history of ICU hospitalization, the presence of urinary catheter, mechanical ventilation, and prior antibiotic exposure [19]. Similarly, another study in the United States developed a score to differentiate BSIs caused by carbapenem-resistant Enterobacteriaceae from those caused by ESBL-producing Enterobacteriaceae, highlighting the significance of specific threshold scores and their predictive power [20]. A study with 337 patients with hematological malignancy devised a risk score to predict an increased risk for *Stenotrophomonas maltophilia* BSIs to guide early therapy. Their study revealed the key risk factors and their respective weights in predicting the likelihood of such infections, thereby facilitating timely intervention [21]. Most of the studies focusing on carbapenem risk analysis have been specifically tailored for Gram-negative microorganisms. In our study, we established a comprehensive score that is applicable to all Gram-negative microorganisms exhibiting resistance to carbapenem. Notably, our scoring system demonstrated comparable reliability to existing scores in the literature and even outperformed many of them. The overall performance of the developed models in our study, denoted as Model 1, Model 2, and Model 3, indicated a promising ability to predict the risk of carbapenem resistance in patients with hematological malignancies

## 5. Conclusions

Carbapenem resistance in Gram-negative bacterial infections is a critical challenge that significantly increases mortality among patients with hematological malignancies, primarily because standard empirical antibiotic guidelines often fail to cover these resistant pathogens. This study successfully addresses this clinical gap by developing and validating three practical bedside scoring models designed to predict the risk of these infections. These models are constructed from a core set of six significant and easily identifiable clinical risk factors: fluoroquinolone use in the last six months, rectal K. *pneumoniae* colonization, the presence of a urinary catheter, transfer from another hospital, a prior carbapenem-resistant infection, and hospitalization for 30 days or more.

By applying this score, clinicians can make more informed decisions, moving beyond standard protocols to initiate early and appropriate empirical therapy for the patients who need it most. The implementation of this carbapenem resistance score offers a valuable tool to improve the effectiveness of initial treatment, which has the potential to reduce mortality in this vulnerable patient population. Furthermore, the risk factors identified could serve as a basis for developing similar predictive tools in other high-risk settings, such as intensive care units.

## Figures and Tables

**Table 1 idr-17-00092-t001:** Comparison of patients with carbapenem-resistant and susceptible GNBIs diagnosed with hematological malignancies according to demographic data, hospitalization, referral status from another hospital or service, hematological disease, stem cell transplantation status and comorbid diseases; results of univariate and multivariate analysis.

Patient Characteristic	Carbapenem Resistant GNBIs (*n* = 64)	Carbapenem Susceptible GNBIs (*n* = 90)		*p*	Multivariate Analysis *p* OR (95% CI)
Age in years (median ± SD)	49.3 ± 16.1	52.2 ± 13.0		0.232	
Sex, female	34 (53.1)	48 (53.3)		0.980	
Hematology Clinic HSCT Clinic	38 (59.4) 26 (40.6)	40 (44.4) 50 (55.6)		0.068	
Acute Leukemia	38 (59.4)	50 (55.6)		0.637	
Lymphoma	16 (25.0)	22 (24.4)		0.937	
Multiple Myeloma	3 (4.7)	8 (8.9)			
Aplastic Anemia	4 (6.3)	7 (7.8)			
Myelodysplastic Syndrome	1 (1.6)	2 (2.2)			
HSCT	21 (32.8)	36 (40.0)		0.363	
Graft-versus-Host Disease	4 (6.3)	12 (13.3)		0.156	
Referral status from another hospital or clinic	10 (15.6)	5 (5.6)		**0.038**	**0.046** 3.148 (1.021–9.711)
Presence of neutropenia during GNBIs	49 (76.6)	65 (72.2)		0.545	
Duration of neutropenia before GNBIs	8.00 (3.50–17.50)	9.00 (3.00–15.50)		0.989	
≥7 days of neutropenia	28 (57.1)	36 (55.4)		0.851	
≥14 days of neutropenia	15 (30.6)	20 (30.8)		0.986	
**Antibiotics used in the last 6 months before GNBIs**					
B-lactam/B-lactamase inhibitor		63 (98.4)	73 (81.1)	**0.001**	**0.010** 14.671 (1.899–113.368)
Aminoglycoside		51 (79.7)	59 (65.5)	0.056	
Fluoroquinolone		48 (75.0)	49 (54.4)	**0.009**	**0.010** 2.510 (1.245–5.063)
Carbapenem		47 (73.4)	42 (46.7)	**0.001**	**0.001** 3.160 (1.581–6.314)
Trimethoprim/Sulfamethoxazole		33 (51.6)	40 (44.4)	0.383	
Colistin 3rd-generation cephalosporin	25 (39.1) 12 (18.7)	12 (13.3) 17 (18.9)		**<0.001** 0.983	**<0.001** 4.167 (1.894–9.166)
Other cephalosporins		3 (4.7)	6(6.7)	0.736	
Use of more than 3 antibiotics		64 (71.1)	57(89.1)	**0.007**	**0.010** 3.308 (1.335–8.199)
The last 6 months before GNBIs Hospitalization history History of carbapenem resistant infection ICU hospitalization history	55 (85.9) 19 (29.7) 19 (29.7)	70 (77.8) 4 (4.4) 11 (12.2)		0.202 **<0.001** **0.007**	**0.001** 9.078 (2.912–28.297) **0.009** 3.032 (1.325–6.939)
Rectal *K. pneumoniae* colonization during GNBIs	39 (60.9)	11 (12.2)		**<0.001**	**<0.001** 11.204 (5.003–25.090)
Steroid treatment before GNBIs	56 (87.5)	74 (82.2)		0.374	
Cumulative dose of methylprednisolone (mg/kg)	13.00 (7.00–22.00)	7.90 (3.65–16.50)		**0.021**	0.227 1.009 (0.995–1.023)
Blood products (last three months)					
Platelets (U)	10.00 (4.00–19.00)	5.00 (2.00–13.25)		**0.023**	0.066 1.023 (0.999–1.048)
Erythrocyte (U)	6.00 (3.00–12.00)	3.00 (2.00–5.00)		**<0.001**	**0.002** 1.164 (1.058–1.280)
Fresh-frozen plasma (U)	4.00 (2.00–13.00)	2.50 (2.00–6.00)		0.216	

Data are expressed as the arithmetic mean ± standard deviation or *n* (%). Important results are in bold. GNBIs: Gram-negative bacterial infections; HSCT: hematopoietic stem cell transplantation; ICU: intensive care unit; *n*: number; OR: odd ratio; CI: confidence interval; SD: standard deviation; *p*: *p* value.

**Table 2 idr-17-00092-t002:** Comparison of antibiotics and chemotherapeutics used in the last 6 months and other risk factors for carbapenem resistance in patients with carbapenem-resistant and susceptible GNBIs diagnosed with hematological malignancy; univariate and multivariate analysis results.

Patient Characteristic	Carbapenem Resistant GNBIs (*n* = 64)	Carbapenem Susceptible GNBIs (*n* = 90)	Total (*n* = 154)	*p*	Multivariate Analysis *p* OR (95% CI)
**Length of hospital stay before GNBIs**	29.50 (17.25–46.50)	19.00 (12.00–26.75)	22.00 (14.00–38.00)	**<0.001**	
≥14 days of stay	56 (87.5)	61 (67.8)	117 (76.0)	**0.005**	**0.006 3.328 (1.404–7.885)**
≥21 days of stay	46 (71.9)	44 (48.9)	90 (58.4)	**0.004**	**0.005 2.672 (1.348–5.294)**
≥30 days of stay	32 (50.0)	21 (23.3)	53 (34.4)	**0.001**	**0.001 3.286 (1.645–6.563)**
**Invasive procedures or surgery**					
Central venous catheter	61 (95.3)	85 (94.4)	146 (94.8)	1.000	
Urinary catheter	22 (34.4)	12 (13.3)	34 (22.1)	**0.002**	**0.003 3.405 (1.534–7.556)**
Hemodialysis	6 (9.4)	5 (5.6)	11 (7.1)	0.527	
Drain catheter	3 (4.7)	1 (1.1)	4 (2.6)	0.308	
Surgical	8 (12.5)	8 (8.9)	16 (10.4)	0.469	
Bronchoscopy	7 (10.9)	4 (4.4)	11 (7.1)	0.202	
Colonoscopy	3 (4.7)	4 (4.4)	7 (4.5)	1.000	
Endoscopy	1 (1.6)	2 (2.2)	3 (1.9)	1.000	

Data are expressed as arithmetic mean ± standard deviation or *n* (%). Important results are in bold. GNBIs: Gram-negative bacterial infections; ICU: intensive care unit; *n*: number; OR; odds ratio; CI: confidence interval; SD: standard deviation; *p*: *p* value.

**Table 3 idr-17-00092-t003:** Calculation of carbapenem resistance risk score in patients with hematological malignancies.

Variables	Coefficient (β)	Standard Error	Wald χ^2^	*p*
**Model 1**				
Fluoroquinolone use in the last 6 months	1.321	0.461	8.219	**0.004**
Rectal *K.pneumoniae* colonization during GNBIs	2.427	0.448	29.374	**<0.001**
Transferred from another hospital or service	1.244	0.634	3.845	**0.050**
Presence of urinary catheter	0.999	0.486	4.220	**0.040**
**Model 2**				
Fluoroquinolone use in the last 6 months	1.110	0.453	5.885	**0.015**
Rectal *K.pneumoniae* colonization during GNBIs	2.325	0.448	26.949	**<0.001**
Presence of urinary catheter	1.086	0.498	4.761	**0.029**
Carbapenem resistant infection in the last 6 months	0.819	0.420	3.799	0.051
**Model 3**				
Fluoroquinolone use in the last 6 months	1.258	0.454	7.676	**0.006**
Rectal *K.pneumoniae* colonization during GNBIs	2.247	0.449	25.087	**<0.001**
Presence of urinary catheter	1.075	0.496	4.702	**0.030**
≥30 days of hospitalization	0.814	0.422	3.723	0.054

Model 1: carbapenem resistance risk score = 1.321 × (fluoroquinolone use in the last 6 months) + 2.427 × (rectal *K. pneumoniae* colonization during GNBIs) + 0.999 × (presence of urinary catheter) + 1.244 × (transferred from another hospital or service) (Hosmer–Lemeshow χ^2^ = 3.309, *p* = 0.652; Cox-Snell R2 = 0.314). Model 1: carbapenem resistance risk score (rounded) = 1.5 × (fluoroquinolone use in the last 6 months) + 2.5 × (rectal *K. pneumoniae* colonization during GNBIs) + 1 × (presence of urinary catheter) + 1 × (transferred from another hospital or service). Model 2: carbapenem resistance risk score = 1.110 × (fluoroquinolone use in the last 6 months) + 2.325 × (rectal *K. pneumoniae* colonization during GNBIs) + 1.086 × (presence of urinary catheter + 0.819 × (infection with carbapenem-resistant in the last 6 months)) (Hosmer–Lemeshow χ2 = 5.703 *p* = 0.575; Cox-Snell R2 = 0.313). Model 2: carbapenem resistance risk score (rounded) = 1 × (fluoroquinolone use in the last 6 months) + 2.5 × (rectal *K. pneumoniae* colonization during GNBIs) + 1 × (presence of urinary catheter) + 1 × (carbapenem resistant infection in the last 6 months). Model 3: Carbapenem resistance risk score = 1.258 × (fluoroquinolone use in the last 6 months) + 2.247 × (rectal *K. pneumoniae* colonization during GNBIs) + 1.075 × (presence of urinary catheter + 0.814 × (30 days or more hospitalization) (Hosmer–Lemeshow χ^2^ = 4.439, *p* = 0.618; Cox-Snell R2 = 0.312). Model 3: carbapenem resistance risk score (rounded) = 1.5 × (fluoroquinolone use in the last 6 months) + 2.5 × (rectal *K. pneumoniae* colonization during GNBIs) + 1 × (presence of urinary catheter) + 1 × (30 days or more hospitalization). Data are expressed as the arithmetic mean ± standard deviation or *n* (%). Important results are in bold. GNBIs: Gram-negative bacterial infections; *n*: number, *p*: *p* value.

## Data Availability

Data available on request.

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
