# Peer review of "Bedside Risk Scoring for Carbapenem-Resistant Gram-Negative Bacterial Infections in Patients with Hematological Malignancies"

_2036-7449, 2025, doi:10.3390/idr17040092_

Round 1

Reviewer 1 Report

Comments and Suggestions for Authors

A Bedside Risk Scoring for Carbapenem-Resistant Gram-Negative Bacterial Infections in Patients with Hematological Malignancies

GNBIs are very common in our modern world, and making difficult the empirical therapies of them.

The topic itself is very actual and interesting, but the manuscript seems a bit chaotic to me. The methodology is not clear enough, not everything is defined properly, and the results and discussion do not follow each other logically, making it very difficult to follow what is written in the manuscript. Moreover, the conclusion is not entirely supported by the results.

Abstract:

The results part does not provide sufficient information about the essential findings.

Introduction:

Is clear and well prepared.

Materials and Methods:

Settings: please give examples for haematological malignancies.

Study design

2.2.1. two times (Lines 60 and 65) is written the study included patients. The two sentences should be combined; it would be much more transparent who was involved in the study.

2.2.2 lines 52 and 70: Only those patients were included who developed the infection in the hospital during the data collection? Or those who had previous hospitalization and had hospital acquired GNBI? Please clarify it! It would be helpful for the reader a chart with inclusion exclusion criteria.

Line 71: It would be helpful a supplementary table with the prednisone equivalent doses.

Lines 74 and 79: “if the pathogen suspected..” which pathogen? Gram negative?

Line 89: “(p)<0.05” why is p in bracket? In other parts of the manuscript is p only. It should be used consistently

I miss the data collection paragraph, listing the data (e.g. demographic, clinical characteristics, comorbidities, lab tests..)

Lines 92-94: I miss the definition of the models in this section. It is not clear. “All three scoring models were constructed using four variables common to each. Scores between 1 and 2.5 were assigned by rounding the coefficient (β) values for the four remaining variables.” I thing it is not enough for the readers.

Results

Line 116: 60 patients retrospectively. It is not defined in the methods why did you make both retrospective and prospective data collection. Was the sample number not enough? Or what is the reason? It should be clarifying in the methods.

Line 130: the abbreviation CRIs is the first time, and it is without explanation. Please verify all abbreviations in the manuscript.

Table 1-2: The column “Total GNBIs” should be cut since it makes no sense, because the purpose of the study was to compare the 2 groups. In fact, it is confusing for the reader if it remains in it. (By the way, this was not specifically stated in the methodology. I deduced it from the introduction.)

Please verify the tables:

e.g. Table 1: p-values and  Multivariate analysis p OR (95% CI) are missing in some of the lines, and I think the lines are shifted here and there (e.g colistin).

The tables contain data that are not listed and defined in the methods. e.g. steroid use status before GNBIs what does it mean?

Or blood products? What does it mean? It is not clear for the readers. Do you mean what was administered to the patients? There are not units for platelets, etc...

HCST under the Table 1 is present, but in the table I cannot find it.

Line 190: “. Important results are in bold. 19” Table 3: there are no results in bold.

Line 199: ROC, AUC are not explained as abbreviations, there are missing from the methods.. It is confusing that the models are only listed here and not in the methodology.

Discussion

My main problem is there is no logical order between the main objectives, results, and discussion.

Please use consistently quinolones or fluoroquinolones.(e.g. Lines 255-259, etc.)

Line 140: “a greater number of erythrocyte and platelet suspensions before the..” Although this attracts the reader's attention, I cannot see the discussion on it in the discussion section

Line 146: “Based on the results of antibiotic susceptibility tests, effective empirical treatment was not 146 initiated in 42% of patients with CRIs.” I would be interested in whether the clinical outcomes would differ between those who had effective empirical treatment and those who had not.

Lines 301-303: “Our study found that hospitalization for 30 days or more before infection and a history of ICUs hospitalization in the last six months tripled the risk of developing carbapenem-resistant GNBIs.” It is not highlighted in the result section. However, it is important to be discussed.

Conclusion

Lines 352-354: “Consequently, the empirical treatment options of- 352 ten involve older agents like colistin and fosfomycin, or relatively newer active agents 353 such as ceftazidime-avibactam, plazomycin, eravacycline, and cefiderocol.” I do not thing that this is the conclusion. It is not supported by the results and discussion.

Here you should highlight the main objectives results.

I appreciate your effort in this study, the work is valuable. However, the manuscript should be revised to make it more readable and the information presented logically and in a way that is understandable to everyone.

Author Response

Comments 1: The results part does not provide sufficient information about the essential findings.

Response 1: Thank you for this observation. The Results section has been substantially revised to better highlight the essential findings of our study.

Introduction:

Is clear and well prepared.

Materials and Methods:

Comments 2:Settings: please give examples for haematological malignancies.

Response 2: As suggested, we have added specific examples of the included hematological malignancies (such as acute leukemia, lymphoma, etc.) to the Methods section for enhanced clarity

Study design

Comments 3: 2.2.1. two times (Lines 60 and 65) is written the study included patients. The two sentences should be combined; it would be much more transparent who was involved in the study.

Response 3: We agree. This paragraph has been edited for improved clarity and flow, combining the sentences to avoid repetition.

Comments 4: 2.2.2 lines 52 and 70: Only those patients were included who developed the infection in the hospital during the data collection? Or those who had previous hospitalization and had hospital acquired GNBI? Please clarify it! It would be helpful for the reader a chart with inclusion exclusion criteria.

Response 4: We have clarified the patient inclusion criteria within the Methods section to specify that the study focuses on hospital-acquired GNBIs. We have also added a detailed data collection paragraph to address this further.

Comments 5:Line 71: It would be helpful a supplementary table with the prednisone equivalent doses.

Response 5: A Supplementary Table detailing the prednisone equivalent doses has been created and referenced in the manuscript.

Comments 6: Lines 74 and 79: “if the pathogen suspected..” which pathogen? Gram negative?

Response 6: The sentence has been revised to explicitly state "Gram-negative microorganism" to remove any ambiguity.

Comments 7: Line 89: “(p)<0.05” why is p in bracket? In other parts of the manuscript is p only. It should be used consistently

Response 7: Thank you for noticing this inconsistency. We have corrected the manuscript to ensure the 'p-value' notation is used consistently throughout.

Comments 8: I miss the data collection paragraph, listing the data (e.g. demographic, clinical characteristics, comorbidities, lab tests..)

Response 8: We have now included a dedicated paragraph under section 2.2.1 detailing the collected data, including demographic, clinical, and comorbidity information

Comments 9:Lines 92-94: I miss the definition of the models in this section. It is not clear. “All three scoring models were constructed using four variables common to each. Scores between 1 and 2.5 were assigned by rounding the coefficient (β) values for the four remaining variables.” I thing it is not enough for the readers.

Response 9: We have expanded upon the model creation process in the Methods section to provide a clearer and more detailed explanation for the reader.

Results

Comments 10:Line 116: 60 patients retrospectively. It is not defined in the methods why did you make both retrospective and prospective data collection. Was the sample number not enough? Or what is the reason? It should be clarifying in the methods.

Response 10: An explanation has been added to the Methods section, clarifying that the prospective data collection phase was initiated to achieve a sufficient sample size for robust statistical analysis.

Comments 11: Line 130: the abbreviation CRIs is the first time, and it is without explanation. Please verify all abbreviations in the manuscript.

Response 11: All abbreviations throughout the manuscript have been carefully reviewed and defined upon their first use.

Comments 12:Table 1-2: The column “Total GNBIs” should be cut since it makes no sense, because the purpose of the study was to compare the 2 groups. In fact, it is confusing for the reader if it remains in it. (By the way, this was not specifically stated in the methodology. I deduced it from the introduction.)

Response 12: We agree that this column could be confusing. In line with your suggestion, the 'Total' column has been removed from the relevant tables to focus the comparison directly on the two study groups.

Please verify the tables:

Comments 13: e.g. Table 1: p-values and  Multivariate analysis p OR (95% CI) are missing in some of the lines, and I think the lines are shifted here and there (e.g colistin).

Response 13:  The tables have been carefully corrected to include the missing data and fix any alignment issues.

Comments 14:The tables contain data that are not listed and defined in the methods. e.g. steroid use status before GNBIs what does it mean?Or blood products? What does it mean? It is not clear for the readers. Do you mean what was administered to the patients? There are not units for platelets, etc...

Response 14: Clearer definitions for these variables, including the units for blood products, have been added to the Methods section to ensure reader understanding.

Comments 15: HCST under the Table 1 is present, but in the table I cannot find it.

Response 15: The abbreviation has now been correctly incorporated into Table 1.

Comments 16: Line 190: “. Important results are in bold. 19” Table 3: there are no results in bold.

Response 16: The important results in Table 3 have now been bolded for emphasis, as intended.

Comments 17: Line 199: ROC, AUC are not explained as abbreviations, there are missing from the methods.. It is confusing that the models are only listed here and not in the methodology.

Response 17: We have moved the detailed information on model generation to the Methods section and ensured all statistical abbreviations, including ROC and AUC, are clearly defined

Discussion

Comments 18:My main problem is there is no logical order between the main objectives, results, and discussion.

Response 18: The discussion section has been revised accordingly.

Comments 20:Please use consistently quinolones or fluoroquinolones.(e.g. Lines 255-259, etc.)

Response 20: The terminology has been revised for consistency; 'fluoroquinolones' is now used throughout the manuscript.

Comments 21: Line 140: “a greater number of erythrocyte and platelet suspensions before the..” Although this attracts the reader's attention, I cannot see the discussion on it in the discussion section

Response 21: Due to the large number of variables analyzed, we chose to focus the Discussion section primarily on the risk factors that were ultimately included in the final scoring models.

Comments 22: Line 146: “Based on the results of antibiotic susceptibility tests, effective empirical treatment was not 146 initiated in 42% of patients with CRIs.” I would be interested in whether the clinical outcomes would differ between those who had effective empirical treatment and those who had not.

Response 22: While this is an interesting clinical question, a comparative analysis of outcomes based on the effectiveness of empirical treatment was beyond the primary scope of this study. To maintain focus, we have removed the sentence that prompted this question.

Conclusion

Comments 23: Lines 352-354: “Consequently, the empirical treatment options of- 352 ten involve older agents like colistin and fosfomycin, or relatively newer active agents 353 such as ceftazidime-avibactam, plazomycin, eravacycline, and cefiderocol.” I do not thing that this is the conclusion. It is not supported by the results and discussion.

Here you should highlight the main objectives results.

Response 23: The Conclusion has been rewritten to remove statements not directly supported by our findings and to better highlight the main results and implications of our study.

Reviewer 2 Report

Comments and Suggestions for Authors

This work is interesting but I think that most of the data presented are already known, concerning the risk factors for  carbapenem-resistance infection, and although they presented a model tu determine a cut-off, the study is restricted to their experience and single center.

The work, if deeply reviewed, could be presented as letter to editor with the presentation/suggestion of a carbapenem-score only.

Table 4 could be submitted as supplementary material

Author Response

Comments 1: This work is interesting but I think that most of the data presented are already known, concerning the risk factors for  carbapenem-resistance infection, and although they presented a model tu determine a cut-off, the study is restricted to their experience and single center.

The work, if deeply reviewed, could be presented as letter to editor with the presentation/suggestion of a carbapenem-score only.

Table 4 could be submitted as supplementary material

Response 1: First and foremost, we would like to thank you for your valuable time and constructive comments on our study. The reviewer's feedback has been instrumental in guiding us to revise and strengthen our manuscript. Accordingly, we have thoroughly revised the manuscript from start to finish.

The reviewer noted that the risk factors for carbapenem resistance presented in our study are known in the literature and questioned the study's novelty. On this point, we would like to respectfully clarify that the primary novel aspect of our work is not the identification of individual risk factors, but rather the development of a quantitative scoring model that combines these factors, specifically within a highly specific and vulnerable patient group: those with hematological malignancies. Our literature search revealed that studies on risk scoring focused on this particular patient population are quite rare. We believe our study fills this gap and therefore provides a significant contribution to the literature.

To more clearly highlight this unique contribution, we have revised the Introduction and Discussion sections to emphasize that the primary aim of our study was to develop a practical clinical scoring tool using these known risk factors.

We agree with the reviewer's valuable suggestion to present Table 4 as supplementary material. Accordingly, Table 4 has now been uploaded as a 'Supplementary Table' in the revised manuscript. We believe this change improves the flow and readability of the main text.

In conclusion, considering that our study focuses on a topic rarely addressed in the literature and given the detailed methodology required for our novel scoring model, we believe that the 'original research article' format is the most appropriate for this work. In light of your feedback, we have resubmitted the thoroughly revised manuscript as an 'original article'.

We hope that in its revised and strengthened form, the manuscript will now be found suitable for publication.

Thank you once again for your valuable contributions.

Reviewer 3 Report

Comments and Suggestions for Authors

The article is highly relevant from a clinical-microbiological perspective, as it addresses the impact of hematological diseases on colonization and infection by carbapenem-resistant strains. The originality of the study is noteworthy, as few comparative studies have been conducted between colonized and non-colonized populations with this type of pathogen, especially in patients with hematological malignancies.

I consider the analysis presented in Table 2 to be highly pertinent; however, it would be helpful to clarify the presentation of the results of the multivariate analysis, particularly the p-values and odds ratios (ORs), as their representation is currently insufficiently explained.

Regarding Table 3 ("Calculation of the risk index for carbapenem resistance in patients with hematological malignancies"), which is one of the most innovative elements of the manuscript, the explanation in the table's footnote is unclear, especially with regard to the different models applied. Likewise, Table 4 creates some confusion, given that its content is neither adequately described nor referenced in the results section.

In this regard, I suggest that these tables (3 and 4) be better integrated into the body of the manuscript, either by explicitly incorporating them into the results section or, if the authors prefer not to overload the main text, by including them as supplementary material duly cited in the discussion.

Finally, while the discussion is adequate, it could be further strengthened by comparing the findings with studies conducted in patients with other non-oncological conditions, which would better contextualize the applicability of the results and broaden their clinical scope.

Author Response

Comments 1:I consider the analysis presented in Table 2 to be highly pertinent; however, it would be helpful to clarify the presentation of the results of the multivariate analysis, particularly the p-values and odds ratios (ORs), as their representation is currently insufficiently explained.

Response-1: Thank you for this comment. We have revised Table 2 to ensure the results of the multivariate analysis, including p-values and odds ratios (ORs), are presented more clearly.

Comments 2:Regarding Table 3 ("Calculation of the risk index for carbapenem resistance in patients with hematological malignancies"), which is one of the most innovative elements of the manuscript, the explanation in the table's footnote is unclear, especially with regard to the different models applied. Likewise, Table 4 creates some confusion, given that its content is neither adequately described nor referenced in the results section.

In this regard, I suggest that these tables (3 and 4) be better integrated into the body of the manuscript, either by explicitly incorporating them into the results section or, if the authors prefer not to overload the main text, by including them as supplementary material duly cited in the discussion.

Response 2: We appreciate the reviewer's feedback on improving the clarity of our tables. To address this, the methodology for the score calculation in Table 3 has been explained in greater detail within the Materials and Methods section. While Table 3 remains in the main text as it forms the backbone of our study, we have moved Table 4 to the supplementary materials as suggested, to streamline the manuscript

Round 2

Reviewer 1 Report

Comments and Suggestions for Authors

The manuscript has been improved.

No other questions.

Reviewer 2 Report

Comments and Suggestions for Authors

The authors revised and corrected the entire article, improving it and highligthing the aim of their article.

I thank the authors and I think that the article is suitable for publication in this form.